# Fluorescence-based index from the Sofia *Streptococcus pneumoniae* fluorescent immunoassay: a prognostic tool for pneumococcal community-acquired pneumonia

Adrián Antuori,[1] Pablo Gonzalez,[2] Alba Llop,[1] Pablo Pillado,[1] María Dolores Quesada,[1] Montserrat Giménez,[1,3] Pere-Joan Cardona[1,3,4]

**ABSTRACT**   Urinary antigen tests, such as the Sofia *Streptococcus pneumoniae* fluorescent immunoassay, provide rapid and specific detection of *S. pneumoniae* in community-acquired pneumonia. Although the assay is qualitative, it generates a semi-quantitative fluorescence-based index whose prognostic utility remains underexplored. This study evaluates the relationship between the Sofia *S. pneumoniae* index and clinical severity in pneumococcal community-acquired pneumonia (CAP). A cross-sectional study of 161 adults with pneumococcal CAP was conducted at a tertiary hospital. Patients with severe pneumonia (quick sequential organ failure assessment (qSOFA) ≥ 2: 10.5 vs 2.3 relative light unit (RLU); CURB-65 ≥ 3: 12.3 vs 2.3 RLU), ICU admission (12.1 vs 2.7 RLU), and 30-day mortality (14.0 vs 3.3 RLU) showed significantly higher median index values ($P < 0.001$). Area under the curves from receiver operating characteristic curve analysis were 0.75 for qSOFA ≥ 2, 0.79 for CURB-65 ≥ 3, 0.71 for ICU admission, and 0.79 for 30-day mortality. Thresholds identified for CURB-65 ≥ 3 prediction were 6.5 RLU (sensitivity: 79.2%; specificity: 75.9%), 4.2 RLU for qSOFA ≥ 2 (sensitivity: 80.4%; specificity: 64.0%), 12.1 RLU for ICU admission (sensitivity: 51.5%; specificity: 81.9%), and 12.7 RLU for 30-day mortality (sensitivity: 78.6%; specificity: 81.5%). A 10-point increase in the index was associated with higher odds of severe pneumonia (CURB-65 odds ratio [OR]: 1.95; qSOFA OR: 1.48), ICU admission (OR: 1.35), and 30-day mortality (OR: 1.42). The Sofia *S. pneumoniae* fluorescent index demonstrates significant prognostic value in pneumococcal CAP. It complements traditional severity scores, offering rapid, microbiologically precise insights for guiding management and identifying high-risk patients.

**IMPORTANCE**  Community-acquired pneumonia caused by *Streptococcus pneumoniae* remains a leading cause of severe illness and death worldwide. Rapid and reliable tools for predicting disease severity are critical to improving patient outcomes. This study evaluates the Sofia *S. pneumoniae* fluorescent immunoassay, a novel urinary antigen test that provides a semi-quantitative fluorescence index associated with severity/outcomes. By linking a fluorescence-based index to clinical outcomes such as ICU admission and mortality, this research demonstrates the potential of graded microbiological signals to complement traditional scoring systems. These findings highlight a valuable step toward integrating microbiological data into clinical decision-making, ultimately enhancing patient care.

**KEYWORDS**  community-acquired pneumonia, *Streptococcus pneumoniae*, urinary antigen tests, prognostic, severity

Address correspondence to Adrián Antuori, aantuori.germanstrias@gencat.cat.

The authors declare no conflict of interest.

*S*treptococcus pneumoniae significantly contributes to morbidity and mortality from community-acquired pneumonia (CAP), particularly in vulnerable populations (1, 2).

Prompt, accurate diagnosis is vital, as timely antibiotic treatment improves outcomes (3). However, conventional methods like cultures have limitations, including low sensitivity and delayed results (4, 5).

Urinary antigen tests (UATs), notably the Sofia *S. pneumoniae* fluorescent immuno-assay (FIA), provide rapid, specific, and non-invasive detection unaffected by prior antibiotic use (6, 7). These tests target the C-polysaccharide antigen, demonstrating approximately 80% sensitivity in CAP cases (8, 9). The automated Sofia FIA employs fluorescence to yield objective, highly sensitive, and specific results (10, 11).

Although primarily qualitative, the Sofia FIA also generates a semi-quantitative, luminescence-based index. While bacterial load correlates with pneumonia severity and outcomes (12), the prognostic value of antigen-based semi-quantitative indices remains insufficiently explored. This study examines the association between the Sofia FIA fluorescence index and clinical severity, including 30-day mortality, in pneumococcal CAP.

## MATERIALS AND METHODS

### Study design

We conducted a cross-sectional study of 161 adult patients (≥18 years) admitted with pneumococcal pneumonia to Germans Trias i Pujol University Hospital (Badalona, Spain), serving a population of 1.4 million, from January 2023 to December 2024. Diagnosis required radiographic evidence of pneumonia plus a positive urinary antigen test (UAT) for *S. pneumoniae* (Sofia *S. pneumoniae* FIA; Quidel, San Diego, USA), or *S. pneumoniae* detection/isolation by one of the following methods: molecular detection from lower respiratory tract specimens (Seegene Allplex PneumoBacter–based workflow), respiratory culture, or blood culture. Patients without radiographic pneumonia or available Sofia antigen results were excluded. Data collected within 48 h of admission encompassed sociodemographic characteristics, clinical indicators, laboratory parameters, and Sofia assay results. Clinical severity and 30-day mortality were evaluated in relation to the Sofia FIA fluorescence index.

All patients were systematically tested for severe acute respiratory syndrome coronavirus 2 (SARS-CoV-2) and influenza at presentation. Cases with a positive SARS-CoV-2 or influenza virus result within 48 h of admission were excluded to avoid mixed viral-bacterial infections. We also excluded bacterial co-infections when a non-pneumococcal bacterial pathogen was detected in lower respiratory tract specimens by respiratory culture and/or our PCR panel (includes *Bordetella parapertussis*, *Bordetella pertussis*, *Chlamydia pneumoniae*, *Haemophilus influenzae*, *Legionella pneumophila*, and *Mycoplasma pneumoniae*, in addition to *Streptococcus pneumoniae*). Because culture and PCR were not performed for all patients, co-infection exclusion relied on the available lower respiratory tract specimen microbiology results.

### Data collection

Clinical data included demographics, pneumonia-related risk factors, CURB-65, quick sequential organ failure assessment (qSOFA) scores, biochemical parameters (C-reactive protein and leukocyte counts), and microbiological cultures (blood and respiratory samples). Urine samples for the Sofia FIA were refrigerated immediately and analyzed within 3 h post-collection.

The Sofia *S. pneumoniae* FIA (Quidel, San Diego, USA) is a qualitative lateral flow immunofluorescence assay designed to classify samples as positive (≥1 relative light unit [RLU]) or negative (−1). Additionally, the assay provides a numerical fluorescence index expressed in relative light units (RLUs), representing the luminescence intensity ratio relative to the predefined assay cut-off value. For statistical analyses and plotting, we assigned randomized values drawn from a uniform (0, 1) distribution to negative results in order to preserve their qualitative nature while allowing rank-based comparisons and

visualization on a logarithmic scale. Data on hospital admissions, ICU admissions, and 30-day mortality were collected and pseudo-anonymized.

## Definitions

Pneumococcal pneumonia was defined as new radiographic infiltrates accompanied by clinical symptoms (fever, cough, dyspnea, and chest pain) and positive culture and molecular detection from lower respiratory tract specimens (Seegene Allplex Pneumo-Bacter–based workflow) or UAT for *S. pneumoniae*. Severe pneumonia was categorized by qSOFA ≥ 2 or CURB-65 ≥3 (13).

## Statistical analysis

We conducted statistical analyses using Python (version 3.9), through libraries such as pandas, numpy, statsmodels, matplotlib, seaborn, SciPy, and scikit-learn. All statistical tests were two-sided, and a *P*-value < 0.05 was considered statistically significant. Because RLU values were right-skewed, we used a logarithmic *y*-axis in scatterplots illustrating the relationship between the Sofia index and severity scores (CURB-65 and qSOFA) to improve visualization across the full dynamic range. Boxplots comparing index distributions across clinical outcomes were presented on the linear scale. All hypothesis testing was performed on raw RLU values using non-parametric methods.

### Descriptive statistics

Descriptive statistics were summarized using medians and interquartile ranges (IQRs) for continuous variables and frequencies and percentages for categorical variables. To evaluate associations between the Sofia *S. pneumoniae* index and clinical or demographic factors, Mann-Whitney *U* tests were employed for binary categorical variables (e.g., gender and comorbidities), and Spearman correlation coefficients were calculated for continuous variables, such as age.

### Linearity and dilution analysis of the Sofia FIA fluorescence index

To assess whether the Sofia fluorescence-based index decreases consistently with progressive dilution (supporting semi-quantitative behavior), we performed a 1:2 serial dilution series on three Sofia-positive urine samples using a Sofia-negative urine matrix as diluent. Dilution factors were 1, 0.5, 0.25, 0.125, 0.0625, and 0.03125. Each dilution was tested in triplicate. Mean ± standard deviation (SD) RLUs were plotted against dilution factor, and monotonic association between dilution factor and mean RLU was assessed using Spearman's correlation.

### Repeatability and reproducibility assessment of the Sofia S. pneumoniae index

We assessed repeatability and reproducibility of the Sofia FIA index using three urine samples representing low (≥1 and ≤5 RLU), medium (>5 and ≤20 RLU), and high (>20 RLU) index values. We tested each sample 10 times per day over 3 consecutive days. A small number of measurements were missing, resulting in 85 observations.

Precision is reported in RLU units as follows: (i) within-day variability, expressed as the standard deviation (SD) of replicate measurements and the corresponding repeatability limit (r), defined as the expected 95% maximum absolute difference between two measurements obtained under repeatability conditions on the same day; and (ii) between-day variability, expressed as the change in daily mean RLU and the corresponding reproducibility limit (R), defined as the expected 95% maximum absolute difference between two measurements obtained on different days. Repeatability and reproducibility limits were estimated using a two-way variance components model with sample and day included as factors.

### Relationship between Sofia S. pneumoniae index values and clinical outcomes

The relationship between Sofia *S. pneumoniae* index values and key clinical outcomes (mortality and pneumonia severity) was assessed uniformly across all analyses. Comparisons between groups (e.g., survivors vs non-survivors, ICU admission vs no ICU admission, and mild vs severe pneumonia) employed the Mann-Whitney *U* test for non-parametric data. Spearman correlations between RLU and CURB-65/qSOFA were computed using available pairs.

To evaluate whether Sofia index values differed according to the type of microbiological documentation obtained in routine clinical care, we compared RLU values between positive and non-positive results using the Mann-Whitney *U* test. Analyses were performed separately for respiratory culture, blood culture, respiratory PCR, and a composite variable capturing any respiratory or blood detection or growth of *S. pneumoniae*.

For bacteremic cases (blood culture–positive pneumococcal CAP), we retrieved serotype data from routine microbiology records and summarized descriptively. Exploratory comparisons of Sofia RLU values across serotypes were limited to serotypes represented by at least two isolates and were performed using the Kruskal-Wallis test.

### Evaluation of prognostic accuracy and optimal thresholds

We generated receiver operating characteristic (ROC) curves to assess the discriminative ability of the Sofia *S. pneumoniae* index for various outcomes, including pneumonia severity (qSOFA ≥ 2 and CURB-65 ≥ 3), ICU admission, and 30-day mortality. Area under the curve (AUC) values were quantified with their 95% confidence intervals (CIs). Furthermore, we compared the ability of Sofia *S. pneumoniae* index, CURB-65, and qSOFA scores for two outcomes, ICU admission, and 30-day mortality. The AUC comparisons were performed using DeLong's test. Optimal thresholds for the Sofia *S. pneumoniae* index were determined for all outcomes (pneumonia severity, ICU admission, and 30-day mortality) using the Youden Index to maximize sensitivity and specificity. Finally, we performed logistic regression analyses to calculate odds ratios (ORs) for associations between increments in the Sofia index and clinical outcomes, with 95% confidence intervals determined using the Wald method.

## RESULTS

### Population characteristics

The study included 161 participants (median age 74 years, IQR: 61–82). Demographic, patient characteristics, laboratory, and outcomes data are summarized in Table 1. Males accounted for 41.6%. Key comorbidities included chronic obstructive pulmonary disease (COPD) (32.3%), cardiovascular disease (36.0%), and type II diabetes mellitus (23.6%).

Hospitalization was required in 88.2%, with 20.5% needing ICU admission. The 30-day mortality rate was 8.7%. Severe pneumonia occurred in 29.2% by qSOFA ≥ 2 and 29.8% by CURB-65 ≥ 3. Respiratory cultures were obtained in 92 of 161 patients (57.1%), with *S. pneumoniae* isolated in 57 patients, corresponding to 35.4% of the total cohort. Blood cultures were obtained in 108 of 161 patients (67.1%), with *S. pneumoniae* bacteremia documented in 31 patients (19.3% of the total cohort). Respiratory PCR testing was available in 46 of 161 patients (28.6%) and was positive for *S. pneumoniae* in 22 patients, representing 13.7% of the total cohort. Using the composite variable capturing any respiratory or blood detection/growth of *S. pneumoniae*, 88/161 patients (54.7%) had documented microbiological detection/growth.

Serotype information was available for all bacteremic episodes (31/31), comprising 14 pneumococcal serotypes. The most frequent serotypes were 3 (*n* = 6), 8 (*n* = 6), and 9N (*n* = 4); serotypes 4, 10A, 12F, and 33F accounted for two isolates each, and the remaining serotypes were singletons (11A, 13, 16F, 17F, 22F, 31, and non-capsulated). In an exploratory comparison restricted to serotypes represented by ≥2 isolates (*n* = 28), Sofia RLU distributions did not differ across serotypes (*P* = 0.313).

**TABLE 1** Sociodemographic, clinical, and outcome profile of patients with pneumococcal community-acquired pneumonia

| Socio-demographic characteristics | (N = 161) |
| --- | --- |
| Age (years) | 74 (61–82) |
| Male gender | 67 (41.6%) |
| Medical history | |
| Antibiotic use within last month | 5 (3.1%) |
| Comorbidities | |
| Smoking | 32 (19.9%) |
| Cardiovascular diseases[a] | 58 (36.0%) |
| Diabetes mellitus 2 | 38 (23.6%) |
| Chronic kidney failure | 24 (14.9%) |
| Chronic obstructive pulmonary disease | 52 (32.3%) |
| Asthma | 16 (9.9%) |
| HIV | 9 (5.6%) |
| Neoplasia | 26 (16.1%) |
| Solid organ transplantation | 0 (0.0%) |
| Cirrhosis | 5 (3.1%) |
| Vital signs at admission[b] | |
| CURB-65 | 2.0 (1.0–3.0) |
| qSOFA | 1.0 (0.0–2.0) |
| CURB-65 ≥ 3 | 48 (29.8%) |
| qSOFA ≥ 2 | 47 (29.2%) |
| Laboratory parameters at admission[b] | |
| Positive urine antigen for *Streptococcus pneumoniae* | 139 (86.3%) |
| Sofia *S. pneumoniae* index (RLU) | 4.1 (1.3–11.9) |
| *Streptococcus pneumoniae* growth in a respiratory sample | 57 (35.4%) |
| *Streptococcus pneumoniae* growth in blood culture | 31 (19.3%) |
| *Streptococcus pneumoniae* detection by PCR from respiratory sample | 22 (13.7%) |
| *Streptococcus pneumoniae* detection/growth in respiratory or blood sample | 88 (54.7%) |
| Leukocyte count ($\times 10^9$/L)[b] | 12.1 (8.5–17.0) |
| CRP (mg/L)[b] | 124.5 (57.0–257.5) |
| Outcomes | |
| Hospitalization[b] | 142 (88.2%) |
| Pleural effusion[b] | 10 (6.2%) |
| ICU admission[b] | 33 (20.5%) |
| 30-day mortality | 14 (8.7%) |

[a]Include Cerebrovascular disease, valvular heart disease, peripheral vascular disease, ischemic heart disease, and heart failure.
[b]Within the first 48 h after admission.

To assess cohort homogeneity and to evaluate whether Sofia RLU values differed according to the microbiological method used in routine care, we compared index values across respiratory culture, blood culture, respiratory PCR, and a composite variable capturing any respiratory or blood detection or growth of *S. pneumoniae*.

Among patients with respiratory cultures ($n = 92$), Sofia index values were similar between those with *S. pneumoniae* growth and those without growth (4.5 vs 4.1 RLU; $P = 0.311$). Among patients with blood cultures ($n = 108$), index values did not differ between bacteremic and non-bacteremic cases (5.7 vs 5.1 RLU; $P = 0.852$). Among patients with respiratory PCR results ($n = 46$), values were comparable between PCR-positive and PCR-negative cases (4.2 vs 5.8 RLU; $P = 0.621$). Using the composite detection/growth variable, index values were also similar between groups (4.3 vs 3.5 RLU; $P = 0.476$). All subsequent analyses were conducted in the full cohort ($N = 161$).

The Mann-Whitney $U$ test revealed no statistically significant differences in Sofia *S. pneumoniae* index between groups based on smoking status (3.6 vs 4.1 RLU, $P = 0.579$), male gender (4.7 vs 4.1 RLU, $P = 0.448$), cardiovascular diseases (3.8 vs 4.2 RLU, $P = 0.676$),

chronic kidney failure (median: 4.2 vs 4.1 RLU, $P$ = 0.831), asthma (median: 3.75 vs 4.30 RLU, $P$ = 0.441), neoplasia (median: 3.35 vs 4.20 RLU, $P$ = 0.297), and diabetes mellitus 2 (median: 2.20 vs 4.50 RLU, $P$ = 0.06).

The Spearman correlation analysis showed no significant monotonic relationship between age and the Sofia index ($\rho$ = 0.073, $P$ = 0.357).

### Effect of serial dilution on the Sofia index

Across three Sofia-positive urine samples, the Sofia fluorescence-based index decreased progressively with 1:2 serial dilution, demonstrating a monotonic relationship between dilution factor and mean RLU (pooled $\rho$ = 0.94, $P$ < 0.001; per-sample $\rho$ = 1.00, $P$ < 0.001 for each sample) (Fig. S1).

### Repeatability and reproducibility metrics

All precision results are expressed in RLUs. We evaluated precision using three urine samples with low, medium, and high Sofia index values. Each sample was measured 10 times per day over 3 consecutive days, resulting in 85 measurements because of occasional missing readings.

Within-day ranged from 0.14 to 0.27 RLU for low values, from 0.24 to 0.41 RLU for medium values, and from 1.6 to 3.1 RLU for high values. Day-to-day changes in the daily mean were small for the low and medium samples, with maximum changes of 0.47 RLU and 0.23 RLU, respectively, and larger for the high sample, with a maximum change of 3.8 RLU. In terms of absolute differences, the estimated repeatability limit was $r$ = 3.9 RLU and the estimated reproducibility limit was $R$ = 4.2 RLU.

### Relationship between Sofia *S. pneumoniae* index and clinical outcomes

The median Sofia index value was significantly higher in patients with severe pneumonia, as determined by qSOFA ≥ 2 (10.5 vs 2.3 RLU, $P$ < 0.001) or CURB-65 ≥ 3 (12.3 vs 2.3 RLU, $P$ < 0.001). Similarly, patients admitted to the ICU (12.1 vs 2.7 RLU, $P$ < 0.001) and those with 30-day mortality (14.0 vs 3.3 RLU, $P$ < 0.001) showed significantly elevated median Sofia *S. pneumoniae* index values (Fig. 1).

Across the full score ranges, the Sofia index showed a positive monotonic association with CURB-65 (Spearman $\rho$ = 0.46, $P$ < 0.001) and qSOFA (Spearman $\rho$ = 0.41, $P$ < 0.001) (Fig. 2).

### ROC curve analysis and optimal threshold determination

For qSOFA-defined severe pneumonia (qSOFA ≥ 2), the AUC of Sofia *S. pneumoniae* index was 0.75 (95% CI: 0.67–0.83, $P$ < 0.001). Similarly, for CURB-65-defined severe pneumonia (CURB-65 ≥ 3), the AUC was 0.79 (95% CI: 0.71–0.87, $P$ < 0.001). The index showed an AUC of 0.71 for ICU admission (95% CI: 0.63–0.80, $P$ = 0.023) and 0.79 for 30-day mortality (95% CI: 0.68–0.88, $P$ = 0.029) (Fig. 3).

For CURB-65-defined severe pneumonia, the associated threshold 6.5 RLU was associated with a sensitivity of 79.2% (95% CI: 66.7%–90.4%) and a specificity of 75.9% (95% CI: 68.1%–83.5%). For qSOFA-defined severe pneumonia, a threshold of 4.2 RLU yielded a sensitivity of 80.4% (95% CI: 69.2%–91.1%) and a specificity of 64.0% (95% CI: 55.1%–72.3%). For ICU admission, a threshold of 12.1 RLU was identified, providing a sensitivity of 51.5% (95% CI: 34.3%–68.0%) and a specificity of 81.9% (95% CI: 74.8%–88.5%). For 30-day mortality, a threshold of 12.7 RLU provided a sensitivity of 78.6% (95% CI: 54.5%–100.0%) and a specificity of 81.5% (95% CI: 75.0%–87.6%).

Furthermore, the predictive accuracy of the Sofia *S. pneumoniae* index, CURB-65, and qSOFA was compared using ROC curve analysis for two outcomes: ICU admission and 30-day mortality. For ICU admission, qSOFA demonstrated the highest AUC (0.85, 95% CI: 0.77–0.91), followed by CURB-65 (0.84, 95% CI: 0.75–0.91), and the Sofia *S. pneumoniae* index (0.71, 95% CI: 0.62–0.80) (Fig. 4). Pairwise comparisons using DeLong's test revealed significant differences between the Sofia *S. pneumoniae* index and both CURB-65 ($P$ =

0.045) and qSOFA (*P* = 0.020), whereas the difference between CURB-65 and qSOFA was not statistically significant (*P* = 0.791).

For 30-day mortality, CURB-65 had the highest AUC (0.90, 95% CI: 0.83–0.96), followed by qSOFA (0.84, 95% CI: 0.72–0.93) and the Sofia *S. pneumoniae* index (0.79, 95% CI: 0.69–0.89) (Fig. 5). DeLong's test comparisons revealed no significant differences among the predictors, with CURB-65 vs the Sofia *S. pneumoniae* index (*P* = 0.075), qSOFA vs the Sofia *S. pneumoniae* index (*P* = 0.536), and CURB-65 vs qSOFA (*P* = 0.363).

## Subgroup analysis in COPD vs non-COPD

Patients with COPD had a lower Sofia index than non-COPD patients (3.15 vs 4.60 RLU, *P* = 0.018), with no differences in severity or outcomes: CURB-65 ≥ 3 (25.5% vs 32.1%, *P* = 0.461), qSOFA ≥ 2 (26.9% vs 30.3%, *P* = 0.714), ICU admission (17.3% vs 22.0%, *P* = 0.538), 30-day mortality (5.8% vs 10.1%, *P* = 0.551), and blood culture positivity (13.5% vs 22.0%, *P* = 0.285).

Of the total study population, 52 participants (32.3%) had COPD. Within this subgroup, the AUC for predicting ICU admission using the Sofia *S. pneumoniae* index was 0.702 (95% CI: 0.514–0.854), compared with 0.695 (95% CI: 0.584–0.806) in the non-COPD group (*n* = 109); DeLong's test indicated no significant difference (*P* = 0.934). For 30-day mortality, the AUC was 0.810 (95% CI: 0.490–1.000) in the COPD group *versus* 0.765 (95% CI: 0.624–0.873) in the non-COPD group (*P* = 0.653). Similarly, there were no significant AUC differences when comparing COPD and non-COPD groups for severe pneumonia defined by CURB-65 ≥ 3 (*P* = 0.505) or qSOFA ≥ 2 (*P* = 0.054).

## Fluorescence index increases and odds of severe outcomes

For a 5-point increase in the Sofia *S. pneumoniae* index, the OR for CURB-65-defined severe pneumonia was 1.40 (95% CI: 1.17–1.66, *P* < 0.001). For qSOFA-defined severe pneumonia, the OR was 1.22 (95% CI: 1.06–1.40, *P* = 0.006); and for ICU admission, the

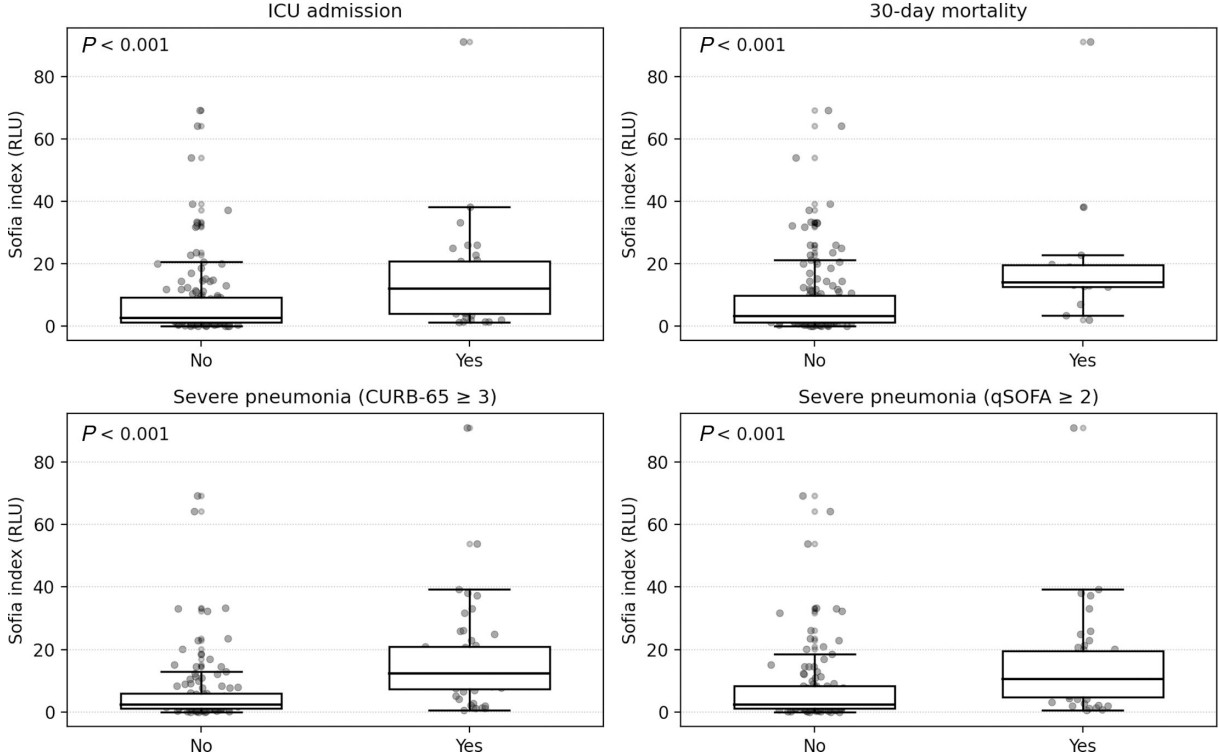

**FIG 1** Sofia index by severe pneumonia and adverse outcomes. Box and whisker plots with overlaid individual values comparing Sofia index distributions by severe pneumonia (CURB-65 ≥ 3 and qSOFA ≥ 2), ICU admission, and 30-day mortality. *P*-values are from two-sided Mann-Whitney *U* tests.

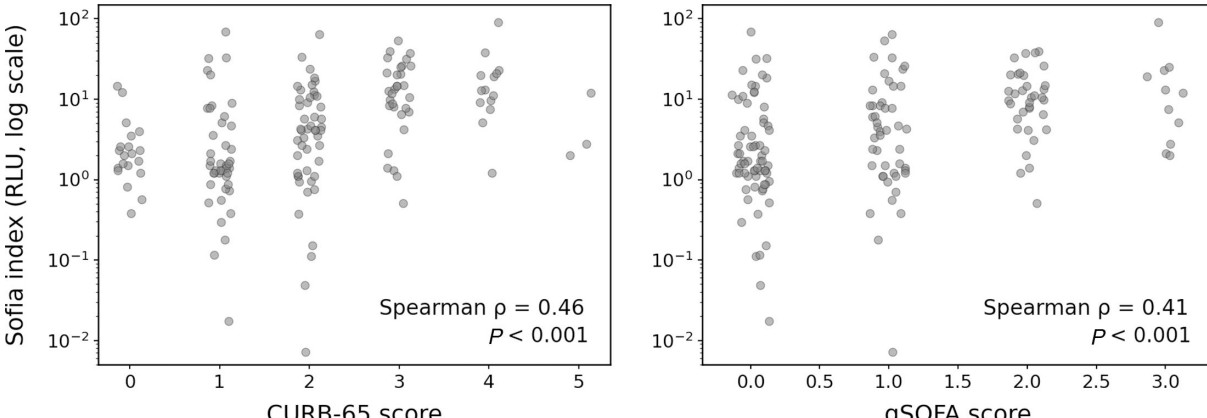

**FIG 2** Sofia index versus severity scores across full ranges (log scale). Scatter plots (with horizontal jitter) of the Sofia *Streptococcus pneumoniae* index (relative light unit [RLU]) *versus* CURB-65 and qSOFA scores. The *y*-axis is logarithmic to improve visualization of the right-skewed RLU distribution. Spearman correlation coefficients and *P*-values are shown in each panel.

OR was 1.17 (95% CI: 1.03–1.34, *P* = 0.016). For 30-day mortality, the OR was 1.24 (95% CI: 1.01–1.52, *P* = 0.037).

For a 10-point increase in the Sofia *S. pneumoniae* index, the OR for CURB-65-defined severe pneumonia was 1.95 (95% CI: 1.38–2.77, *P* < 0.001). For qSOFA-defined severe pneumonia, the OR was 1.48 (95% CI: 1.12–1.95, *P* = 0.006). For ICU admission, the OR was 1.35 (95% CI: 1.05–1.72, *P* = 0.016), and for 30-day mortality, the OR was 1.42 (95% CI: 1.08–1.86, *P* = 0.012) (Fig. 6).

## DISCUSSION

CAP remains a significant cause of morbidity and mortality worldwide, especially among older adults and individuals with underlying health conditions (14). Traditionally, tools such as the Infectious Diseases Society of America/American Thoracic Society criteria (15), CURB-65, the Pneumonia Severity Index (PSI), and sepsis-related tools like qSOFA have been used to assess disease severity and inform treatment decisions (14, 16).

Although these tools provide valuable guidance for managing pneumonia, they also have notable limitations. Their static scoring systems may oversimplify complex clinical scenarios, exhibit age-related bias, and offer limited applicability in

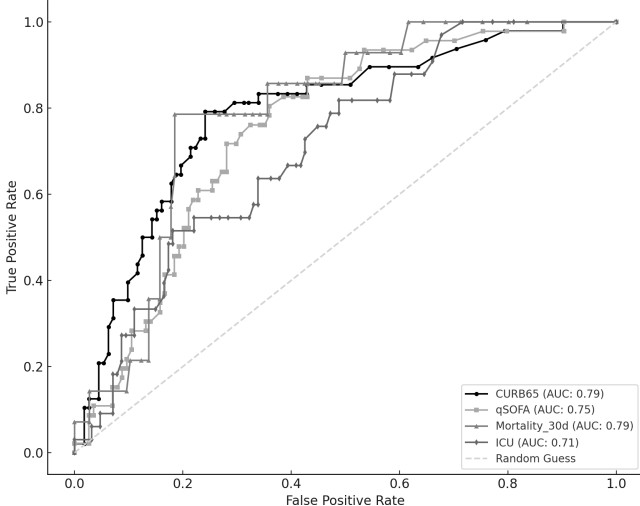

**FIG 3** ROC curve analysis of Sofia *Streptococcus pneumoniae* index for predicting severity, ICU admission, and 30-day mortality in pneumococcal community-acquired pneumonia.

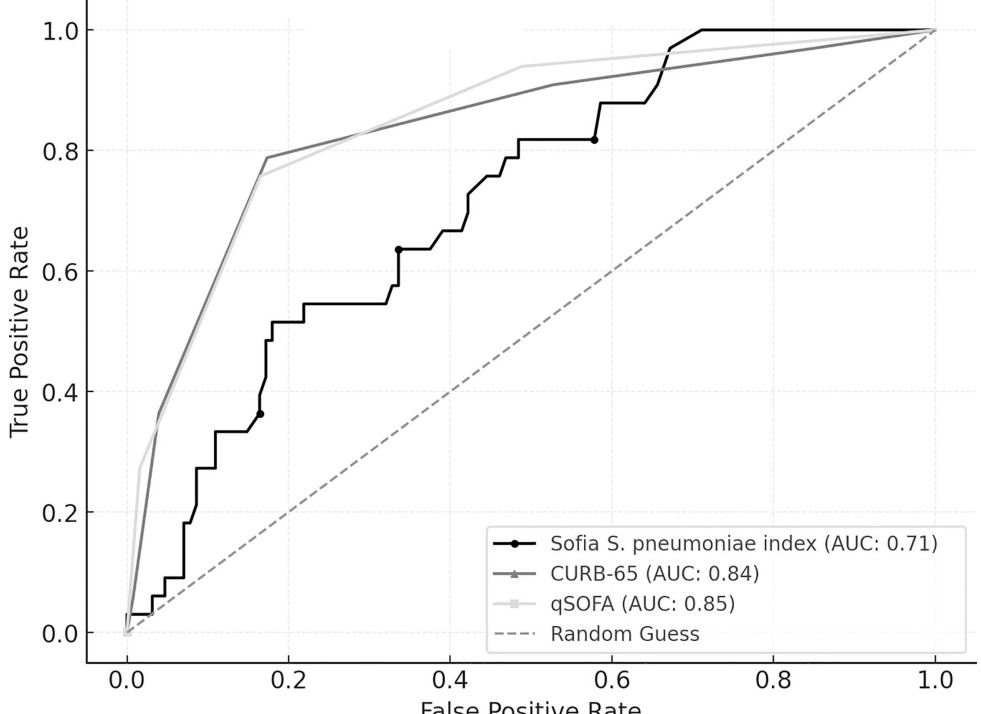

**FIG 4**  Comparison of ROC curve analysis for ICU admission: Sofia *S. pneumoniae* index, CURB-65, and qSOFA.

immunocompromised or atypical cases. Additionally, they focus primarily on mortality risk or ICU admission, overlooking broader management needs (14, 16).

Our results align with earlier studies highlighting the diagnostic capabilities of UATs. Tests such as the Binax NOW test have been implemented for their rapidity and specificity in detecting *S. pneumoniae* (12). UATs have also demonstrated utility in improving etiological identification (17) and reducing the use of broad-spectrum antibiotics (6).

More recently, the Sofia *S. pneumoniae* FIA, fundamentally a qualitative immunofluorescence assay, provides a numerical, fluorescence-based index expressed in RLU. This index reflects signal intensity relative to a predefined positivity threshold rather than absolute quantitative measurement. In our study, this value proved to be a valuable marker to predict *S. pneumoniae* severe pneumonia, providing a novel and user-friendly diagnostic tool. In a serial dilution experiment using three Sofia-positive urine samples, RLU values decreased monotonically with progressive dilution, supporting a semi-quantitative relationship between the Sofia index and urinary pneumococcal antigen burden within the tested range.

Importantly, Sofia RLU values were comparable across patients regardless of whether pneumococcal documentation was obtained by blood culture, respiratory culture, or respiratory PCR. This finding is consistent with a clinically homogeneous pneumococcal CAP cohort in which all cases fulfilled microbiological inclusion criteria, including urinary antigen positivity. It also indicates that the prognostic association of the Sofia index was not driven by the method used to document pneumococcal infection.

Other studies have investigated the impact of quantitative or semi-quantitative indices on disease severity in various conditions, including COVID-19 through viral load (18), invasive aspergillosis using galactomannan levels (19), and cytomegalovirus disease in patients undergoing solid organ transplantation (20). In pneumococcal pneumonia, bacterial load has been proposed as a reliable marker for distinguishing between colonization from active infection (21) and identifying severe disease (12). Rello et al., for instance, demonstrated the association between genomic bacterial load on whole blood

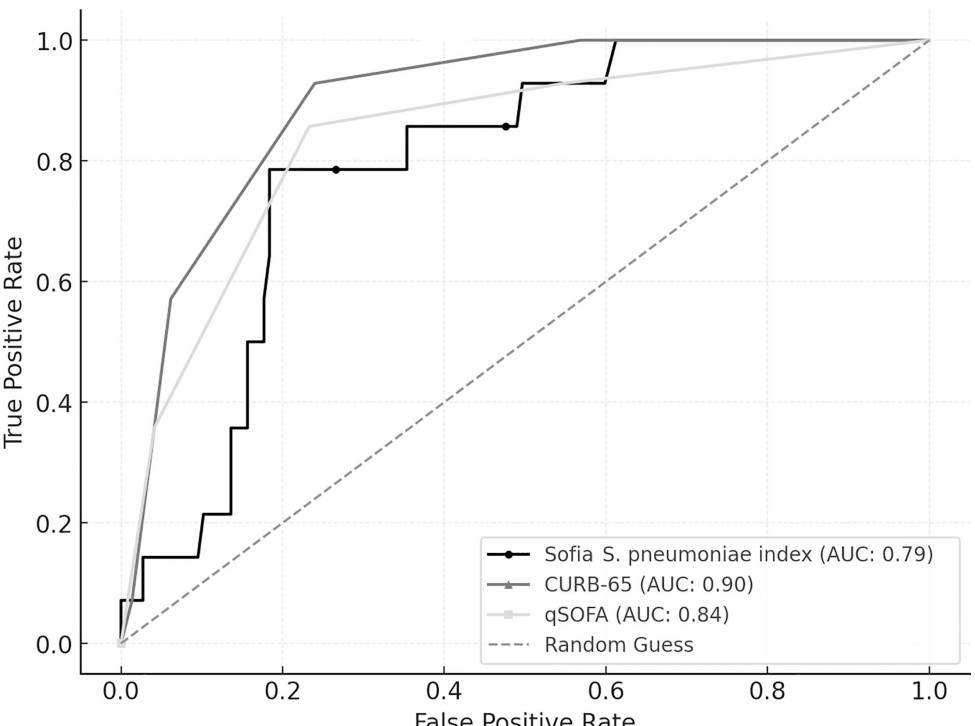

**FIG 5** Comparison of ROC curve analysis for 30-day mortality: Sofia *S. pneumoniae* index, CURB-65, and qSOFA.

and severe outcomes, such as septic shock and mechanical ventilation requirements. In their study, a bacterial load exceeding three logs demonstrated a sensitivity of 66.7% and a specificity of 80% for predicting septic shock (12).

Building on this, antigen-based fluorescence or semi-quantitative indices could bridge the gap between diagnostic speed and prognostic accuracy. The Sofia FIA, which provides a real-time fluorescence-based index, may offer a faster and more accessible alternative to PCR in time-critical settings. Antigen-based techniques, like the Sofia FIA, are less prone to sample degradation, require minimal specialized personnel, and are more feasible in resource-limited environments.

Other biomarkers, such as procalcitonin, have shown prognostic value when tracked over time, with AUC for mortality prediction improving from 0.60 at baseline to 0.73 with serial measurements (22, 23). However, its need for repeat testing and modest early predictive power limits its usefulness in acute decision-making. By contrast, the Sofia index demonstrated comparable AUC values for ICU admission and mortality using a single measurement, simplifying acute care assessments.

In our study, the Sofia *S. pneumoniae* index achieved an AUC of 0.71 for ICU admission, which, although moderate, is slightly lower than the AUC values from CURB-65 and qSOFA scores. Furthermore, for ICU admission, the Sofia index demonstrated a sensitivity of 51.5% and a specificity of 81.9%, comparable to those reported in a validation study of the CURB-65 score (sensitivity of 57.7% and specificity of 64.7% for CURB-65 ≥ 3) (24). However, the discriminatory power for ICU admission of CURB-65 and qSOFA in this retrospective study is considerably higher than in other studies. This difference is likely due to variations in sample size and patient characteristics, given that the present study exclusively included patients with pneumococcal pneumonia (24–26).

For mortality prediction, CURB-65 again showed the best performance (AUC 0.90), followed by qSOFA (AUC 0.84) and the Sofia index (AUC 0.79). However, in this case, these differences were not statistically significant. This is consistent with findings from studies such as Ewig et al., which validated the high accuracy of CURB-65 (24) and qSOFA (27) in mortality prediction among patients with CAP. Although the Sofia index demonstrated

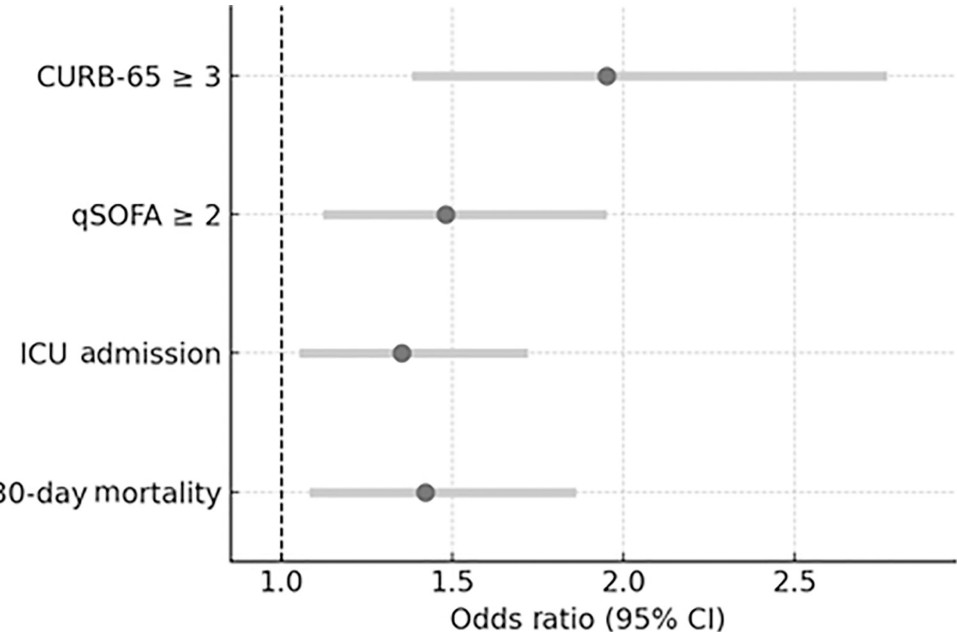

**FIG 6** Odds ratios per 10-point increase in the Sofia *S. pneumoniae* index for clinical severity, ICU admission, and 30-day mortality.

lower discriminative capacity for ICU admission, its performance in predicting mortality was comparable to that of CURB-65 and qSOFA. This suggests its potential utility as an adjunct prognostic tool that contributes microbiological specificity to clinical assessment.

The Sofia technique distinguishes itself as a robust tool for pneumonia risk stratification due to its independence from key demographic and clinical risk factors such as age, smoking status, and comorbidities like chronic kidney failure, asthma, and neoplasia. Severity scores like PSI, CURB-65, and qSOFA are heavily influenced by age and chronic conditions, and they often classify elderly patients into higher risk categories, even when their pneumonia may not be particularly severe (24–26). This independence enhances its applicability across diverse patient populations, which revealed no statistically significant differences in the Sofia index among groups based on these comorbidities, with the exception of COPD, where the index was significantly lower in affected individuals. Although the Sofia index was lower among individuals with COPD, subgroup analyses did not reveal significant differences in its overall performance.

In exploratory subgroup analyses, none of the observed differences between patients with and without COPD reached statistical significance. Patients with COPD tended to have lower Sofia RLU values, whereas markers of disease severity (CURB-65 ≥ 3, qSOFA ≥ 2, ICU admission, and 30-day mortality) and the frequency of bacteremia were consistently higher in patients without COPD. The consistent direction of these findings suggests that non-COPD patients in our cohort may have more often presented with a more severe or invasive clinical phenotype. This could plausibly be associated with higher urinary pneumococcal antigen levels and, consequently, higher Sofia RLU values.

Given the retrospective study design, the limited statistical power for subgroup analyses, and the non-systematic nature of microbiological sampling, these observations should be interpreted as hypothesis-generating. Future prospective studies with larger cohorts may help determine whether different Sofia index cutoffs are needed for patients with COPD.

Importantly, the Sofia index enabled risk stratification based on incremental increases in fluorescence intensity. Higher index values were significantly associated with increased odds of severe pneumonia (as defined by qSOFA and CURB-65), ICU admission,

and 30-day mortality. For each 10-RLU increase, the corresponding odds ratios were 1.95, 1.48, 1.35, and 1.42, respectively. These findings underscore one of the key advantages of the Sofia index: the ability to provide real-time, objective estimates of disease severity independent of demographic or comorbidity-based risk factors.

Together, these findings indicate that the prognostic signal captured by the Sofia index reflects biologically meaningful variation in pneumococcal antigen burden across a clinically homogeneous cohort, rather than differences driven by diagnostic work-up or microbiological sampling.

This study has several limitations. First, although the Sofia FIA is qualitative by design, it displays a fluorescence index in RLUs without standardized concentrations or external calibration, which limits inter-center comparability; our inter-day experiment supports short-term consistency, but RLUs remain a semi-quantitative measure. Second, this was a single-center, retrospective study in a tertiary hospital; despite predefined clinical-radiologic-microbiologic criteria, information bias and limited generalizability cannot be excluded. Third, although bacterial co-infections documented from lower respiratory tract samples were excluded, bacterial PCR and culture were not performed in all patients. By contrast, SARS-CoV-2 and influenza were systematically tested, and positives were excluded. Other respiratory viruses, however, were not evaluated in a standardized manner. Pneumococcal urinary antigen tests may exhibit cross-reactivity with other organisms. Because lower respiratory tract microbiology was not systematically obtained for all patients, we cannot fully exclude that a subset of positive Sofia results (particularly among UAT-only cases) could have been influenced by unrecognized cross-reacting pathogens. Reassuringly, among the limited number of patients in whom non-pneumococcal respiratory pathogens were identified (eight patients), all had a negative Sofia result; however, this does not eliminate the possibility of cross-reactivity in patients without microbiological confirmation. Fourth, serotype information was available only for blood culture isolates and was not systematically available for non-bacteremic cases; therefore, serotype-related analyses were exploratory and limited to bacteremic episodes. Other key variables were not systematically captured, including clinical (obesity), radiologic (imaging patterns), and treatment-related factors (timing and appropriateness of antimicrobials). Fifth, the median age was 74, which may limit applicability to younger populations; external validation in broader age ranges and multi-center settings is warranted. Finally, although the overall sample size supported the primary severity outcomes, subgroup and mortality analyses may be underpowered. Together, these factors should be considered when interpreting the magnitude and generalizability of the associations observed. Prospective, multi-center studies with standardized serotyping, comprehensive co-pathogen panels, protocolized pre-analytics, and serial RLU measurements are needed to refine thresholds and clinical integration.

In summary, while traditional severity scores and other biomarkers offer valuable insights into CAP severity, the Sofia *S. pneumoniae* fluorescence index facilitates the integration of diagnostic speed, microbiological precision, and prognostic accuracy. Its integration into clinical practice could streamline severity assessment, guide timely interventions, and improve outcomes, particularly in time-sensitive or resource-limited settings. Future research should focus on further validating its use across larger cohorts, exploring its role in guiding targeted antimicrobial stewardship programs, and investigating the potential value of serial measurements in monitoring patient evolution.

## ACKNOWLEDGMENTS

Conceptualization: A.A.; data curation: A.A. and P.G.; formal analysis: A.A., P.G., A.L., and P.P.; investigation: A.A., P.G., A.L., P.P., M.G., and P.-J.C.; methodology: A.A.; project administration: A.A.; resources: A.A. and P.G; software: A.A.; supervision : A.A.; validation: A.A.; Visualization: A.A and P.G; writing—original draft: A.A.; writing—review and editing: A.A., P.G., A.L., P.P., M.G., and P.-J.C.

## AUTHOR AFFILIATIONS

[1]Microbiology Department, North Metropolitan Clinical Laboratory, Germans Trias i Pujol University Hospital, Badalona, Spain

[2]Laboratory Medicine Department, North Metropolitan Clinical Laboratory, Germans Trias i Pujol University Hospital, Badalona, Spain

[3]Centro de Investigación Biomédica en Red de Enfermedades Respiratorias (CIBERES), Madrid, Spain

[4]Genetics and Microbiology Department, Universitat Autònoma de Barcelona, Cerdanyola del Vallès, Spain

## AUTHOR ORCIDs

Adrián Antuori http://orcid.org/0000-0002-2782-3125

## AUTHOR CONTRIBUTIONS

Adrián Antuori, Conceptualization, Data curation, Formal analysis, Funding acquisition, Investigation, Methodology, Project administration, Resources, Software, Supervision, Validation, Visualization, Writing – original draft, Writing – review and editing | Pablo Gonzalez, Data curation, Formal analysis, Investigation, Resources, Visualization, Writing – review and editing | Alba Llop, Formal analysis, Investigation, Writing – review and editing | Pablo Pillado, Formal analysis, Investigation, Writing – review and editing | María Dolores Quesada, Resources, Writing – review and editing | Montserrat Giménez, Investigation, Writing – review and editing | Pere-Joan Cardona, Investigation, Writing – review and editing

## DATA AVAILABILITY

The study protocol, statistical analysis plan, and databases are available to anyone who requests them. These requests should be directed to Adrián Antuori (aantuori.germans-trias@gencat.cat).

## ETHICS APPROVAL

The study was conducted according to the principles of the Declaration of Helsinki and was approved by the Ethics Committee of the Germans Trias i Pujol University Hospital (approval application number PI-24-280). The Ethics Committee of the Germans Trias i Pujol University Hospital waived the requirement for informed consent because of the study's retrospective nature, and all data were analyzed anonymously. All methods of the study were carried out in accordance with institutional guidelines and regulations.

## ADDITIONAL FILES

The following material is available online.

### Supplemental Material

**Figure S1 (Spectrum03304-25-S0001.tif).** Serial dilution response of the Sofia *S. pneumoniae* FIA index. Three urine samples were tested in triplicate across a 1:2 serial dilution series. Points represent mean RLU values and error bars indicate ± Standard deviation. The x-axis shows dilution factors (most concentrated at 1, followed by sequential dilutions).
**Supplemental legend (Spectrum03304-25-S0002.docx).** Legend for Figure S1.

## Open Peer Review

**PEER REVIEW HISTORY (review-history.pdf).** An accounting of the reviewer comments and feedback.

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
