## [Reviewer comments · Microbiology Spectrum]

Microbiology Spectrum

Fluorescence-Based Index from the Sofia *S. pneumoniae* FIA: A prognostic tool for pneumococcal community-acquired pneumonia

Adrian Antuori, Pablo Gonzalez, Alba Llopol, Pablo Pillado, María Dolores Quesada, Montserrat Giménez, and Pere-Joan Cardona

Corresponding Author(s): Adrian Antuori, Hospital Universitari Germans Trias i Pujol

Review Timeline:

Submission Date:	October 15, 2025
Editorial Decision:	November 29, 2025
Revision Received:	December 22, 2025
Accepted:	January 26, 2026

Editor: Daniel Ortiz

Reviewer(s): The reviewers have opted to remain anonymous.

Transaction Report:

DOI: <https://doi.org/10.1128/spectrum.03304-25>

Re: Spectrum03304-25 (**Fluorescence-Based Index from the Sofia S. pneumoniae FIA: A prognostic tool for pneumococcal community-acquired pneumonia**)

Dear Dr. Adrian Antuori:

Thank you for the privilege of reviewing your work. Below you will find my comments, instructions from the Spectrum editorial office, and the reviewer comments.

There are many gaps that remain unaddressed after revision. The Sofia S. pneumoniae FIA uses purified rabbit polyclonal antibodies specific to Streptococcus pneumoniae cell wall polysaccharide antigen (CWPS) types 1 and 2 to capture and detect the CWPS antigens. In theory, a higher RLU reading would correspond to a higher bacterial load in the urine. However, the authors have not established that the RLU readings are semi-quantitative with a serial dilution assay. Without this study these findings may just be an anomaly of the patient population studied. Maybe Table 1 should be broken down by RLUs. Is it just that sicker patients with pneumonia tend to shed more S. pneumoniae in the urine? What about a scatter plot of RLUs in relation to the various parameters of severity, ICU Admission, and 30-day mortality?

In addition, pneumococcal urine antigen tests are known to have cross reactivity issues with alpha-hemolytic streptococci and Gram-negative bacteria. Since not all patients had a culture performed, there is potential that some of the higher index values could be attributed to these other cross-reacting organisms instead of the S. pneumoniae.

In terms of bacterial load, culture positive patients would be expected to have higher RLUs than culture negative patients. Were there differences in RLUs in culture positive vs culture negative samples?

Line 66: What was the urinary antigen test used?

Line 142: "Of the total patients, 19.1% had a positive blood culture, 35.2% had a positive respiratory sample culture, and 13.5% had a positive PCR test for S. pneumoniae in respiratory specimens." Does not match data in Table 1.

Any differences noted in RLUs among the different positive results (blood vs. respiratory vs. PCR)?

Do the numbers in the repeatability study represent in RLUs? What was the difference in RLUs?

Revision Guidelines

Publication Fees: For information on publication fees and which article types are subject to charges, visit our website. If your manuscript is accepted for publication and any fees apply, you will be contacted separately about payment during the production

process; please follow the instructions in that e-mail. Arrangements for payment must be made before your article is published.

Sincerely,
Daniel Ortiz
Editor
Microbiology Spectrum

Reviewer #1 (Comments for the Author):

In this study the authors evaluate the relationship between a urine antigen test for *S. pneumoniae* and the clinical severity of pneumococcal respiratory tract infections. The assay used is the Sofia FIA, a qualitative test, that provides an RLU index. The authors use this RLU-index and assess the correlation to disease severity in 161 adult patients with pneumococcal CAP. The study shows interesting findings, but the following questions and comments need to be addressed:

1. Lines 142-143: "Of the total patients, 19.1% had a positive blood culture, 35.2% had a positive respiratory sample culture, and 13.5% had a positive PCR test for *S. pneumoniae* in respiratory specimens".

Is there a correlation between positive blood culture (for pneumococci) and the median Sofia index in this study? If so, the authors should include separate analyses for blood culture positive vs negative patients and the correlation of Sofia RLUs to qSOFA/CURB-65/ICU-admission/mortality for the two different groups.

2. Lines 154-155: "For COPD, the median Sofia index was significantly lower in individuals with COPD (3.15 RLU) compared to those without COPD (4.60 RLU) ($p = 0.018$)."

Could the authors please explain why the Sofia index was lower in the COPD group compared to the group without COPD? Could the authors also include information on the number of patients with positive blood cultures in the two groups?

3. Lines 229-236 regarding bacterial load.

There is a need to show that the RLU correlates with bacterial load. Could the authors include a dilution series with a few urine samples or similar? If not, I think that the authors would need to tone down the discussion section a bit. In addition, I think it is difficult to compare a proxy such as the Sofia FIA for bacterial infections with ct-values for viral infections such as COVID-19 and CMV.

4. As an earlier reviewer pointed out, I think it is important to include some information on pneumococcal serotypes included in this project. Are there any serotype specific variations that may limit the use of this assay as a semi-quantitative analysis?

Response to reviewers

Re: Spectrum03304-25 (Fluorescence-Based Index from the Sofia *S. pneumoniae* FIA: A prognostic tool for pneumococcal community-acquired pneumonia)

Dear Dr. Adrian Antuori:

Thank you for the privilege of reviewing your work. Below you will find my comments, instructions from the Spectrum editorial office, and the reviewer comments.

1. There are many gaps that remain unaddressed after revision. The Sofia *S. pneumoniae* FIA uses purified rabbit polyclonal antibodies specific to *Streptococcus pneumoniae* cell wall polysaccharide antigen (CWPS) types 1 and 2 to capture and detect the CWPS antigens. In theory, a higher RLU reading would correspond to a higher bacterial load in the urine. However, the authors have not established that the RLU readings are semi-quantitative with a serial dilution assay. Without this study these findings may just be an anomaly of the patient population studied. Maybe Table 1 should be broken down by RLUs. Is it just that sicker patients with pneumonia tend to shed more *S. pneumoniae* in the urine? What about a scatter plot of RLUs in relation to the various parameters of severity, ICU Admission, and 30-day mortality?

We agree with the reviewer that an analytical demonstration of semi-quantitative behavior is important. We therefore performed a serial dilution (1:2) linearity experiment on three Sofia-positive urine samples, using a Sofia-negative urine matrix as diluent. Across the dilution series, the Sofia index decreased monotonically with dilution factor, supporting semi-quantitative behavior of the numerical readout (pooled Spearman $\rho = 0.94$, $p < 0.001$; per-sample Spearman $\rho = 1.00$, $p < 0.001$). We have added these data as a new figure (**Supplementary Figure S1**) and updated the Methods, Results, and Discussion sections accordingly.

In addition, to avoid over-interpretation of the Sofia RLU readout as a direct quantitative measure of bacterial load, we revised the manuscript to harmonize terminology. The Sofia output is now consistently described as a semi-quantitative

fluorescence-based index, reflecting signal intensity relative to the assay cut-off and serving as an indirect proxy of urinary antigen burden rather than an absolute quantification.

We agree and have added scatter/box-strip visualizations showing the distribution of Sofia RLUs across severity and outcomes, and the relationship between RLUs and CURB-65/qSOFA across their full score ranges. In the full cohort, RLUs were significantly higher in severe pneumonia by qSOFA ≥ 2 (median 10.5 vs 2.3 RLU, $p < 0.001$) and CURB-65 ≥ 3 (12.3 vs 2.3 RLU, $p < 0.001$), as well as in ICU admission (12.1 vs 2.7 RLU, $p < 0.001$) and 30-day mortality (14.0 vs 3.3 RLU, $p < 0.001$). Across score ranges, RLUs showed a positive monotonic association with CURB-65 (Spearman's $\rho = 0.46$, $p < 0.001$) and qSOFA ($\rho = 0.41$, $p < 0.001$). These plots complement the dichotomized comparisons and ROC analyses by displaying the full distribution and the monotonic trend between increasing clinical severity and higher Sofia index values.

2. In addition, pneumococcal urine antigen tests are known to have cross-reactivity issues with alpha-hemolytic streptococci and Gram-negative bacteria. Since not all patients had a culture performed, there is potential that some of the higher index values could be attributed to these other cross-reacting organisms instead of the *S. pneumoniae*.

We agree with the reviewer that potential cross-reactivity is an inherent limitation of pneumococcal urinary antigen testing, and that our retrospective study was not specifically designed to formally evaluate cross-reactivity against alternative pathogens. Because lower-respiratory-tract microbiological investigations (culture/PCR) were not systematically obtained for all patients, we cannot fully exclude that unrecognized cross-reacting organisms could have influenced a subset of Sofia-positive results (particularly among cases without microbiological confirmation).

To provide context from the available data, eight patients within the cohort had positive respiratory microbiology identifying pathogens other than *S. pneumoniae*; in all

of these cases the Sofia assay result was negative. While this observation is reassuring, it does not eliminate the possibility of cross-reactivity in patients without comprehensive microbiological testing. Accordingly, we explicitly added this point to the Limitations section of the Discussion in the revised manuscript.

3. In terms of bacterial load, culture positive patients would be expected to have higher RLUs than culture negative patients. Were there differences in RLUs in culture positive vs culture negative samples?

We appreciate this important point. In our study, however, “culture-negative” (or PCR-negative) does not imply absence of pneumococcal pneumonia. All included patients fulfilled the criteria for microbiologically confirmed pneumococcal community-acquired pneumonia (CAP) by at least one accepted diagnostic method, namely urinary antigen positivity and/or pneumococcal detection or growth in respiratory or blood samples. Respiratory specimens and blood cultures were not obtained systematically as part of routine care, and information on prior antibiotic exposure was not consistently available. Therefore, a negative result for a given diagnostic modality most likely reflects sampling practices and/or limited test sensitivity rather than true absence of pneumococcal infection.

On this basis, we compared Sofia relative light units (RLUs) between positive and non-positive results for each diagnostic modality among patients with available data. No significant differences were observed: respiratory culture positive vs non-positive (median 4.5 vs 4.1 RLU; $p = 0.311$), blood culture positive vs negative (median 5.7 vs 5.1 RLU; $p = 0.852$), and respiratory PCR positive vs negative (median 4.2 vs 5.9 RLU; $p = 0.621$). Similarly, a composite variable capturing any pneumococcal detection or growth in respiratory or blood samples showed comparable Sofia values (median 4.3 vs 3.5 RLU; $p = 0.476$).

We interpret these findings as supporting the clinical and microbiological homogeneity of the cohort under real-world diagnostic conditions, rather than questioning the

validity of the Sofia quantitative index. We have added these analyses and their rationale to the revised Methods and Results sections.

4. Line 66: What was the urinary antigen test used?

The urinary antigen test used was the Sofia *S. pneumoniae* FIA (Quidel, San Diego, USA). We have now clarified this in the manuscript by revising the sentence at Line 66 to specify the assay:

5. Line 142: "Of the total patients, 19.1% had a positive blood culture, 35.2% had a positive respiratory sample culture, and 13.5% had a positive PCR test for *S. pneumoniae* in respiratory specimens." Does not match data in Table 1.

We reviewed all microbiological denominators and percentages in the manuscript, including Table 1 and the corresponding Results section. Reporting was standardized to use the total study cohort as the denominator throughout (N = 161). After these revisions, all microbiological proportions reported in the Results section are fully consistent with Table 1.

6. Any differences noted in RLUs among the different positive results (blood vs. respiratory vs. PCR)?

This point has already been addressed in our response to the related microbiology-comparison comment above. Briefly, we compared Sofia RLUs across patients with pneumococcal documentation by respiratory culture, blood culture, or respiratory PCR and found no significant differences between modalities (all $p > 0.05$), as now reported in the revised Methods and Results.

7. Do the numbers in the repeatability study represent in RLUs? What was the difference in RLUs?

Yes. All repeatability and reproducibility measurements were reported in relative light units (RLUs). To address the limited interpretability of the previous version, we expanded and clarified the precision study by testing three urine samples spanning low (1–5 RLU), medium (5–20 RLU), and high (>20 RLU) index values. Each sample was measured 10 times per day over three consecutive days.

We additionally reported precision in clinically interpretable RLU units, including within-day standard deviations, maximum day-to-day changes in daily mean values, and the repeatability (r) and reproducibility (R) limits estimated from a two-way variance components model with sample and day as factors.

Reviewer #1 (Comments for the Author):

In this study the authors evaluate the relationship between a urine antigen test for *S. pneumoniae* and the clinical severity of pneumococcal respiratory tract infections. The assay used is the Sofia FIA, a qualitative test, that provides an RLU index. The authors use this RLU-index and assess the correlation to disease severity in 161 adult patients with pneumococcal CAP. The study shows interesting findings, but the following questions and comments need to be addressed:

1. Lines 142-143: "Of the total patients, 19.1% had a positive blood culture, 35.2% had a positive respiratory sample culture, and 13.5% had a positive PCR test for *S. pneumoniae* in respiratory specimens".

Is there a correlation between positive blood culture (for pneumococci) and the median Sofia index in this study? If so, the authors should include separate analyses

for blood culture positive vs negative patients and the correlation of Sofia RLUs to qSOFA/CURB-65/ICU-admission/mortality for the two different groups.

As mentioned in our response to the related comment, in this cohort a “culture-negative” result does not mean absence of pneumococcal pneumonia. All included patients fulfilled criteria for microbiologically confirmed pneumococcal CAP by at least one accepted diagnostic method, and respiratory and blood samples were not collected systematically in routine clinical care.

To further address the reviewer’s concern, we compared Sofia RLUs between positive and non-positive results for each diagnostic modality among patients with available data. No significant differences in RLUs were observed between groups (respiratory culture positive vs non-positive: $p = 0.311$; blood culture positive vs negative: $p = 0.852$; respiratory PCR positive vs negative: $p = 0.621$; composite respiratory or blood detection/growth vs none: $p = 0.476$). These analyses are now reported in the revised Methods and Results sections.

Given the non-systematic nature of culture sampling, and because culture negativity cannot be considered a reliable indicator of lower bacterial burden in this setting (and may be influenced by unmeasured prior antibiotic exposure), we did not perform additional severity or outcome analyses stratified by culture or PCR status. Instead, severity and outcome analyses were conducted in the full cohort.

2. Lines 154-155: "For COPD, the median Sofia index was significantly lower in individuals with COPD (3.15 RLU) compared to those without COPD (4.60 RLU) ($p = 0.018$)."

Could the authors please explain why the Sofia index was lower in the COPD group compared to the group without COPD? Could the authors also include information on the number of patients with positive blood cultures in the two groups?

As requested by the reviewer, we additionally compared severity markers and clinical outcomes between patients with and without COPD. We did not observe statistically significant differences in CURB-65 ≥ 3 , qSOFA ≥ 2 , ICU admission, or 30-day mortality (all $p > 0.05$). However, these indicators were consistently more frequent in the non-COPD group, which could plausibly have contributed to higher RLU values. We emphasized that this interpretation was hypothesis-generating and not intended as a definitive explanation.

Regarding bacteremia, we compared blood culture positivity between groups and observed positivity in 7/52 patients (13.5%) in the COPD group and in 24/109 patients (22.0%) in the non-COPD group ($p = 0.285$). We added these comparisons and the above clarification to the revised Results and Discussion sections.

3. Lines 229-236 regarding bacterial load.

There is a need to show that the RLU correlates with bacterial load. Could the authors include a dilution series with a few urine samples or similar? If not, I think that the authors would need to tone down the discussion section a bit. In addition, I think it is difficult to compare a proxy such as the Sofia FIA for bacterial infections with ct-values for viral infections such as COVID-19 and CMV.

We agree that RLUs should not be interpreted as a direct measure of bacterial load. To address the request for experimental support of semi-quantitative behavior, we performed a 1:2 serial dilution series on three Sofia-positive urine samples using a Sofia-negative urine matrix as diluent (dilution factors 1 to 0.03125; triplicate testing). Across all three samples, RLUs decreased monotonically with progressive dilution, with strong rank correlation between dilution factor and mean RLU (per-sample Spearman $\rho = 1.00$, $p < 0.001$; pooled $\rho = 0.94$, $p < 0.001$), supporting that the Sofia index behaves semi-quantitatively with respect to urinary pneumococcal antigen burden.

To reflect this, we have explicitly strengthened the Limitations section to state that the Sofia FIA is qualitative by design and reports an RLU fluorescence index without

standardized concentrations or external calibration, which limits inter-center comparability; while our inter-day experiment supports short-term consistency, RLUs should be regarded as a semi-quantitative measure.

4. As an earlier reviewer pointed out, I think it is important to include some information on pneumococcal serotypes included in this project. Are there any serotype specific variations that may limit the use of this assay as a semi-quantitative analysis?

We agree that serotype-related variation is an important consideration when interpreting a semi-quantitative urinary antigen index. We therefore compiled pneumococcal serotype information for all bacteremic episodes in our cohort (31/31 blood culture-positive cases). These cases included 14 distinct serotypes, most frequently serotypes 3 and 8, followed by 9N. In an exploratory analysis restricted to serotypes represented by at least two isolates ($n = 28$), Sofia RLU distributions did not differ across serotypes ($p = 0.313$). These data have been added to the Results. We also clarified in the Limitations that serotyping was available only for blood culture isolates. Larger prospective studies with systematic serotyping in both bacteremic and non-bacteremic pneumococcal CAP would therefore be required to more definitively assess serotype-specific effects on the Sofia index.

Re: Spectrum03304-25R1 (**Fluorescence-Based Index from the Sofia S. pneumoniae FIA: A prognostic tool for pneumococcal community-acquired pneumonia**)

Dear Dr. Adrian Antuori:

Your manuscript has been accepted, and I am forwarding it to the ASM production staff for publication. Your paper will first be checked to make sure all elements meet the technical requirements. ASM staff will contact you if anything needs to be revised before copyediting and production can begin. Otherwise, you will be notified when your proofs are ready to be viewed.

Sincerely,
Daniel Ortiz
Editor
Microbiology Spectrum